# Uninterrupted HIV treatment for women: Policies and practices for care transitions during pregnancy and breastfeeding in Côte d'Ivoire, Lesotho and Malawi

Tamsin K. Phillips[1]*, Halli Olsen[2], Chloe A. Teasdale[2,3,4], Amanda Geller[5], Mamorapeli Ts'oeu[6], Nicole Buono[7], Dumbani Kayira[7], Bernadette Ngeno[5], Surbhi Modi[5], Elaine J. Abrams[2,3,8]

1 Division of Epidemiology and Biostatistics, School of Public Health & Family Medicine, University of Cape Town, Cape Town, South Africa, 2 Mailman School of Public Health, ICAP-Columbia University, New York, New York, United States of America, 3 Department of Epidemiology, Mailman School of Public Health, New York, New York, United States of America, 4 Department of Epidemiology and Biostatistics, CUNY Graduate School of Public Health & Health Policy, New York, New York, United States of America, 5 Division of Global HIV & TB, US Centers for Disease Control and Prevention (CDC), Atlanta, Georgia, United States of America, 6 Division of Global HIV & TB, CDC-Lesotho, Maseru, Lesotho, 7 Division of Global HIV & TB, CDC-Malawi, Lilongwe, Malawi, 8 College Physicians and Surgeons, Columbia University, New York, New York, United States of America

* tammy.phillips@uct.ac.za

**Data Availability Statement:** All relevant data are within the paper and its Supporting information files. The full transcripts from this study are not

## Abstract

Transitions between services for continued antiretroviral treatment (ART) during and after pregnancy are a commonly overlooked aspect of the HIV care cascade, but ineffective transitions can lead to poor health outcomes for women and their children. In this qualitative study, we conducted interviews with 15 key stakeholders from Ministries of Health along with PEPFAR-supported and other in-country non-governmental organizations actively engaged in national programming for adult HIV care and prevention of mother-to-child-transmission of HIV (PMTCT) services in Côte d'Ivoire, Lesotho and Malawi. We aimed to understand perspectives regarding transitions into and out of PMTCT services for continued ART. Thematic analysis revealed that, although transitions of care are necessary and a potential point of loss from ART care in all three countries, there is a lack of clear guidance on transition approach and no formal way of monitoring transition between services. Several opportunities were identified to monitor and strengthen transitions of care for continued ART along the PMTCT cascade.

## Introduction

While substantial progress has been made towards prevention of mother-to-child transmission of HIV (PMTCT) over the past two decades, loss to follow-up of women along the PMTCT cascade is still common with implications for HIV infections in children and for maternal health [1]. As part of universal antiretroviral therapy (ART), all people living with HIV should

publicly available as approval for public dissemination of the raw transcripts was not obtained as it may still be possible to identify participants through their transcripts. Transcripts may be available upon request by contacting the Columbia University Institutional Review Board (IRB) at +1 212-305-5883 or irboffice@columbia.edu.

**Funding:** The authors listed from ICAP at Columbia University received funding from the US Centers for Disease Control and Prevention (U2GGH00994) to complete the work described in the paper. All funding was received by Columbia University (no individual funding). TKP was supported by a VECD Global Health Fellowship, funded by the Office of AIDS Research (OAR) and the Fogarty International Center (FIC) of the NIH (D43 TW009337). The findings and conclusions in this report are those of the authors and do not necessarily represent the official position of the funding agencies. The funders had no role in study design, data collection and analysis, decision to publish, or preparation of the manuscript.

**Competing interests:** The authors have declared that no competing interests exist.

receive lifelong ART for their own health and prevention of HIV transmission [2]. In 2019, there were an estimated 150,000 new HIV infections in children, the majority resulting from in utero transmissions related to mothers not receiving ART during pregnancy (29%) and mothers dropping out of ART care during pregnancy (12%). In addition, mothers not receiving ART or dropping out of ART care during breastfeeding contributed to 13% and 11% of all new HIV infections in children, respectively [3, 4].

One aspect of the cascade that is commonly overlooked, and which may contribute to women not receiving ART or discontinuing ART, is the transition between different services to continue treatment during and after pregnancy. Successful transition between services for ART is not well defined but is considered here as completing the step of moving from one service and entering a new service. In African studies this ranges from between 25% and 67% during pregnancy [5–7] and from 74% to 77% postpartum [8–10]. Although specific service delivery models for HIV care during and after pregnancy differ across Africa, transition between different clinical services is required in many countries [11–14]. Fig 1, from a recent systematic review [15], illustrates three points of maternal transition for ongoing HIV treatment which present threats to uninterrupted HIV treatment along the PMTCT cascade. These transition points are defined as: 1) women on ART who become pregnant and need to transition into antenatal care (ANC) and PMTCT services, 2) pregnant women who are not on ART and need to transition into integrated ART and ANC (i.e., services offered in the same clinic and during the same visits) or stand-alone ANC and ART services, and 3) women who received integrated ART and ANC during pregnancy and need to transition to stand-alone ART services after delivery.

The aim of this qualitative study was to understand the perspectives and experiences of key stakeholders from three PMTCT priority countries regarding how women living with HIV are transitioned into and out of PMTCT services for continued ART [16]. These findings provide new information to inform policymakers and key stakeholders on the existing landscape of transition practices and challenges, as well as opportunities for strengthening transitions along the PMTCT cascade.

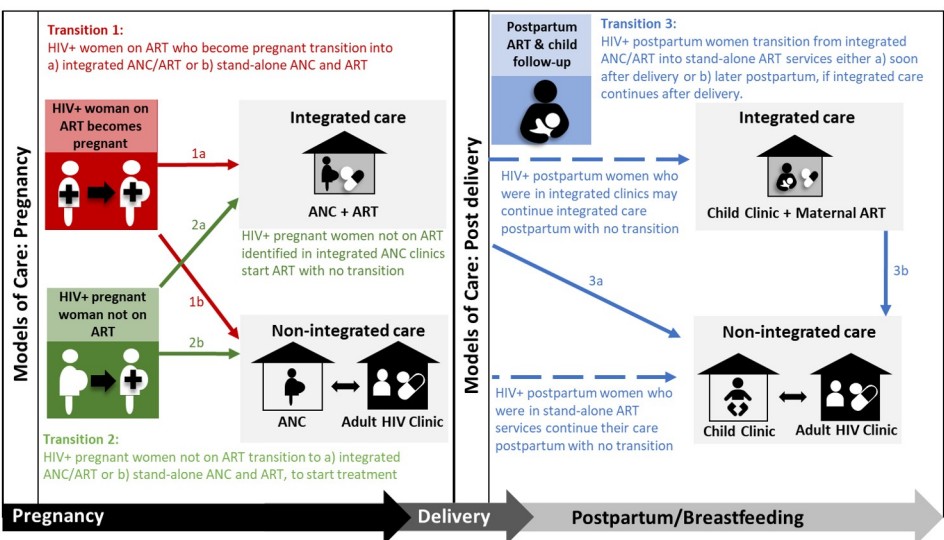

**Fig 1. Points of transition into and out of PMTCT for continued ART during and after pregnancy.** ART—antiretroviral therapy; ANC—antenatal care. Reprinted from Phillips TK et al., J Int AIDS Soc 2020, 23:e25633, original copyright CC BY 4.0 2020.

## Methods

Key informant interviews (KII) were conducted in three countries: Côte d'Ivoire, Lesotho and Malawi. These countries were selected by ICAP, in collaboration with the US Centers for Disease Control and Prevention (CDC), based on a guideline review and the following criteria: (1) being one of the 21 Global Plan high-priority PMTCT countries in sub Saharan Africa [17]; (2) having country guidance either related to the transition of HIV-positive women to and from PMTCT services or that described where women living with HIV should receive ANC and ART care during and after pregnancy; and (3) having a CDC country office. The three countries were selected based on the above criteria as well as in-country ethical board submission requirements and timelines (see information below on approvals).

### Setting and country guidelines

While the adult HIV prevalence varies in Côte d'Ivoire (2.4%), Lesotho (22.8%) and Malawi (8.9%), all three are priority PMTCT countries with final vertical transmission rates in 2020 of 8% in Côte d'Ivoire and 6% in Lesotho and Malawi [18]. Fertility rates are high (4.6, 3.1 and 4.1 births per women in 2019 in Côte d'Ivoire, Lesotho and Malawi, respectively) [19] and in 2020 there were an estimated 15 000, 7 800 and 40 000 births to women living with HIV in Côte d'Ivoire, Lesotho and Malawi, respectively [18]. Malawi has the highest ART coverage in pregnancy (100%) followed by Lesotho (84%) and Côte d'Ivoire (80%) [4].

All three countries have guidance on where women living with HIV should receive ANC and PMTCT services. Broadly, ANC refers to the provision of routine maternity care for all pregnant women. A minimum of eight visits are recommended by the World Health Organization which include maternal and fetal assessments, detection and management of complications and comorbidities, and counselling. PMTCT services refer to the additional care required for women living with HIV and their infants during and after pregnancy to prevent HIV transmission, of which provision of ART is a core component and the focus of this study. Guidelines from all three countries specify that women who are already on ART at conception are referred into ANC and PMTCT services. Integrated ANC and PMTCT care is routinely provided during pregnancy, except for Malawi where the vast majority but not all facilities have integrated ANC and HIV care. The 2016 Lesotho [20] and Côte d'Ivoire [21] national guidelines specify that mothers and babies should receive integrated care in the maternal and child health clinic until 24 and 18 months postpartum, respectively. In Malawi [22] there are three main models of care: 1) the majority of services offer integrated antenatal, HIV, and postpartum care through 24 months postpartum, while some offer 2) integrated antenatal and HIV care with transition to separate postpartum and HIV services at approximately 6 weeks postpartum, or 3) HIV care provided separately from both antenatal and postpartum care. None of the three countries had guidance specific to the process or documentation of transition into or out of PMTCT for continued ART.

### Sampling

A convenience sample of eligible local stakeholders were invited to participate in the KII. Potential participants included staff from the Ministry of Health (MOH), PEPFAR-supported implementing partners, and other in-country non-governmental organizations (NGO). Eligible participants for the KII were those actively engaged in national programming for adult care and PMTCT services (i.e., individuals who contribute to the development of national guidelines either through their job or as a member of a relevant technical working group). Additional eligibility criteria included being aged 18 years or older, willing and able to provide informed consent and able to participate in an interview conducted in English (Lesotho and Malawi) or French (Côte d'Ivoire). ICAP worked closely with the CDC project leads in Atlanta

and with CDC country teams to identify potential key informants (KIs) to participate. The interviewers contacted stakeholders in each country via email using an approved recruitment script to identify interested participants.

## Interviews and analysis

All participants provided written informed consent prior to the interview. Interviews took approximately an hour and were conducted using an approved semi-structured guide of questions that asked about steps in the PMTCT cascade based on the research teams experience and existing literature (S1 Appendix). The KIIs focused predominantly on gathering information about transition steps and related activities in each country. No theoretical framework was used and the interview guide was reviewed by CDC country staff but not piloted. Interviews took place in a private office and were audio recorded and transcribed. Interviews in Malawi (January 2020) and Lesotho (December 2020) were conducted in English; in Côte d'Ivoire (January 2020), interviews were conducted in French and translated and transcribed into English. The interviews in Malawi and Lesotho were conducted during 5-day in-country visits by the ICAP-New York project coordinator who had no existing relationship with the KIs. The Côte d'Ivoire interviews were conducted by the ICAP Country Director in Côte d'Ivoire who did know some of the KIs through their work. Both interviewers had postgraduate research training and prior experience with HIV and PMTCT programs and research. Completed transcripts were not shared with participants for review.

All interview transcripts were coded using Dedoose, a web-based software created by University of California, Los Angeles (UCLA), United States [23]. Interview transcripts were reviewed by one reviewer (HO) and coded into pre-defined concepts guided by a recent systematic review of transitions of care in PMTCT [15]. Identified concepts were grouped into categories and themes uniting the categories that were identified. These categories and themes were discussed and revised with a second reviewer (TP). A summary of common themes and outliers was then drafted along with direct supporting quotes. Draft results were shared with co-authors and in-country CDC staff for feedback and revision. All co-authors have a strong interest in optimizing continuity of care and completion of all steps in the PMTCT cascade with combined programmatic and research expertise. The mix of co-authors with and without experience working in the three included countries provided a balance of perspectives and contextual knowledge in the analysis and interpretation of results.

## Ethical considerations

Ethical approval was obtained from the Columbia University Medical Center (CUMC) Institutional Review Board (IRB), the National Health Sciences Research Committee (NHSRC) in Malawi, the Research and Ethics Committee (REC) in Lesotho and the Comite National D'Ethique des Sciences de la Vie et de la Sante (CNESVS) in Côte d'Ivoire. The protocol was also reviewed in accordance with CDC human research protection procedures and was determined to be non-research.

## Results

There were 33 potential KIs identified (14 in Malawi, 7 in Lesotho and 12 in Côte d'Ivoire). Fifteen stakeholders (five per country) responded to the invitation and agreed to participate, including stakeholders working in the local government at MOH or district level (two from Malawi, two from Lesotho, one from Côte d'Ivoire), PEPFAR-supported implementing partners (three from Malawi, four from Lesotho, one from Côte d'Ivoire) and NGOs (three from Côte d'Ivoire). All participants were directly involved in national programming for PMTCT in

**Table 1. Description of key informant interview participants in Côte d'Ivoire, Lesotho, and Malawi.**

| Characteristic | n (%) |
|---|---|
| **Sector (n = 15)** | |
| NGO | 3 (20) |
| Implementing partner | 8 (53) |
| Government | 4 (27) |
| **Number of years working in current post*** | |
| 0–2 | 1 (7) |
| 3–5 | 4 (29) |
| 6+ | 9 (64) |
| **Sex*** | |
| Female | 12 (86) |
| Male | 2 (14) |
| **Age in years*** | |
| 31–40 | 3 (21) |
| 41–50 | 6 (43) |
| 51–60 | 4 (29) |
| 60+ | 1 (7) |

*n = 14 as one key informant did not provide demographic information.

Malawi, Lesotho, or Côte d'Ivoire (Table 1). Almost all participants were female (86%) and 93% had three or more years of experience in their current post.

While models of care varied and challenges specific to the three countries were described, the emergent themes were very similar across settings. As such, the results are presented as overarching themes with discussion of the specific country experiences within each theme. Transition challenges and opportunities to strengthen transition into and out of PMTCT services for continued ART are discussed (Fig 2).

## Recognition that transitions into and out of PMTCT services for continued ART are required

KIs from all three countries recognized that maternal transitions to continue ART must occur along the PMTCT cascade. Integration of pregnancy, postpartum, child health and HIV services along the cascade has removed much of the need for transitions during pregnancy and early postpartum, but a transition after this integrated period is still required to continue ART in the later postpartum period. Also, integration is not uniformly implemented across all health facilities, so transitions are still needed during pregnancy to receive ART or continue ART in some settings.

"…women benefit from the whole service package during her pregnancy. But what about after her pregnancy? She must either go back to regular adult care pathway or stay"

(KI 5, Côte d'Ivoire)

"In this case, this woman was getting treatment from ART clinic. But when she becomes pregnant, now she starts attending ANC. So, it varies, how you say, there are three models, depending on the infrastructure of the facility and probably preference of how easy it is."

(KI 5, Malawi)

## Challenges identified with transition for continued ART along the PMTCT cascade

- Models of care and infrastructure vary across facilities, resulting in different transition approaches
- There is poor communication, limited documentation and fragmented linkage between transferring and receiving health services
- Women are not traceable through transitions of care for continued ART
- There is no formal handover of responsibility to ensure patient follow up
- Inconsistent partner and community engagement to support transitions of care

## Opportunities identified to address these challenges

**Formalize standard operating procedures**
- Develop standard operating procedures for transitions along the PMTCT cascade which:
    - clearly define transition timing, steps and protocols
    - clearly assign responsibilities to transferring and receiving health services

**Implement monitoring and evaluation tools**
- Implement tools to:
    - facilitate tracing and follow-up through transitions of care
    - enable quantification of women lost to follow-up through a transition
    - enable measurement of the impact of interventions to support transition of care

**Community and partner engagement**
- Strengthen partner engagement and improve links between community and health services to support transitions of care
- Possible strategies include:
    - mentor mothers/peer navigators
    - community-based differentiated models of care
    - education and awareness around antenatal and HIV care services in communities

**Fig 2. Summary of the challenges and opportunities identified in key informant interviews to strengthen transition of care for continued ART along the PMTCT cascade.**

*"Integration of PMTCT, integration of HIV services within the MCH department, maternal and childhood department. Even family planning. Much is on paper, but when it comes to practice, there's just a lot that needs to be done to make sure that there's really good integration of these services."*

*(KI 4, Malawi)*

*". . .and then continue to have care [at ANC] up until when the baby is two-years-old or 24 months old. . .This time when she attends she will be already informed that there will the time that she will be graduated to go to the other care [ART service]."*

*(KI 5, Lesotho)*

Even where integration is implemented in most facilities, such as in Côte d'Ivoire, women do not always stay in one facility.

*"Normally, guidelines state that when a woman tests positive, she must stay within the ANC services until she gives birth and all that. But in some cases, the woman comes and goes. She comes to us only to measure the size of her belly, take her booklet and goes elsewhere to receive her ARV."*

*(KI 5, Côte d'Ivoire)*

Transition into ANC services for women on ART who become pregnant was raised as a challenge in Malawi and Côte d'Ivoire. Specifically lack of pregnancy testing in ART services was thought to delay transition of women into ANC and PMTCT services but active testing and reporting on pregnancies in ART services is not required.

*"I would say that it [diagnosing a pregnancy] is physical or if the woman brings it up on her own. If she has an idea, the woman might say, "I am pregnant" and bring it to the provider's attention. It might be a nurse or someone else. As soon as the women tells her doctor, "I am pregnant," and the provider receives that information, they know that she needs a midwife. So, she is referred to PMTCT."*

*(KI 2, Côte d'Ivoire)*

*"I would say maybe she identifies herself to be pregnant but of course, even when you come in the ART, I think, there are some questions related to [reproductivity] you are asked. So, which of those points if you have, for example, missed your period, then the—now the next step would be actually going to ANC. But by and large, . . .we don't do pregnancy tests in ART as a routine service provision."*

*(KI 5, Malawi)*

### Lack of clear guidance on how health services should support the transition and which provider is responsible for ensuring a woman transitions successfully

Although the need for women to transition between services to continue ART along the PMTCT cascade was acknowledged by all the KIs, they also all spoke to a lack of standard operating procedures and clear guidance on the transition process and responsibilities of the transferring and receiving services. KIs described few formal communication links between the services to ensure that when a woman does have to transition, that this is documented, and she is not lost along the way.

*"We really need support, like the documentation, so that even if we are not around, people can know if I'm working at the main ART, what steps [are needed], if I meet with a mother who is pregnant today, the steps to formal [transfer to PMTCT], and how do I communicate with my colleagues in MCH [maternal and child health]? If they are 24 months [postpartum], how do I communicate with the main ART corner? So, you need that documentation in the referral system."*

*(KI 2, Lesotho)*

*"We don't have them standardized. They have only the improvised. . . I think the transition process should be clearly stipulated in the guidelines so that everybody knows what to do and also do a standard thing in all our facilities."*

(KI 5, Lesotho)

*"So, think we need to have specific SOPs, clearly stating the steps, and the timeframe, people in ANC have to stay with this woman, and the child, and at what point the mother would be transferred back to the mainstream ART. So, we need to have site SOPs."*

(KI 3, Malawi)

*". . .Some procedures have been developed for that. But currently, there are some gray zones."*

(KI 5, Côte d'Ivoire)

In Malawi specifically, KIs reported confusion about when the end of the integrated period is and when women should leave the ANC and go back to the ART clinic. How long women are kept in integrated maternal and child health services postpartum was also influenced by lack of clarity in the guidelines, and varying facility infrastructure and resources.

*"I think the best thing is to clearly indicate the specific time we need to keep a woman within ANC, and the specific time we need to transfer this woman back to main ART, and develop specific counseling guides that we can give to women that are in ANC, when they shall transition to the main ART, and even those that away in main ART. How are we transferring them to ANC to contain within ANC, and at what particular time shall they be transferred back to the main ART?"*

(KI 3, Malawi)

*"But what is recommended is actually you should continue getting treatment [at ANC] for as long as you still breastfeeding. So, some sites will keep you now for six weeks. Some sites will keep for a year. Some sites may keep you for a longer period, depending, like I said, on the personnel, one; two, on infrastructure because there are some sites are just so congested so you are running out to ART clinics within the same facility."*

(KI 5, Malawi)

In Côte d'Ivoire, KIs reported that there is quite clear guidance that women should remain in integrated maternal and child health services until 18 months postpartum, but that there is little guidance on how to approach the transition at 18 months. Often women stay much longer as they have developed relationships with the providers and are hesitant to move to routine HIV services.

*"By the end of those 18 months, they have grown accustomed to the women, to the attention they receive and everything that is done for them, and also the time. You will agree with me that in terms of care, there are enough people who are cared for, so in terms of time, and the fact that they have grown accustomed to the midwives. So those 18 months are not always complied with."*

(KI 2, Côte d'Ivoire)

*"They prefer to stay [in ANC], either because they trust the midwife, or because task shifting [integration] has been well done, they stay with us."*

*(KI 5, Côte d'Ivoire)*

### Lack of monitoring and evaluation tools to track women across transition points

A key challenge reported in all interviews was the inability to track women through transitions of care, to document transitions and to ensure completion of all the steps required for effective PMTCT. They reported that it was not possible to tell whether a woman had in fact linked to the new facility or service after she had been transferred out and that the receiving service may not know to expect her. The most significant implication of this was the inability to quantify loss to follow-up after a transition. With no formal hand over there is also confusion about who is responsible for following up with a woman who should have transitioned. The lack of monitoring and evaluation tools means that links between ANC and ART services are not formalized and rely on individual provider initiative to ensure women have successfully linked to the next step of their care.

*"I think strategies that should be in place should include making sure that there is linkage, if I've transferred out. . . making sure that we follow that specific client up to make sure that they did end up in the facility that we transferred them out to. . .. when we send them back into their main adult ART corner, they do get there and they're linked back into care or they're linked to care."*

*(KI 1, Lesotho)*

*"The only challenge is the M&E [monitoring and evaluation] tools, because we're not using electronic information, which is not connected. So, at clinics that are using paper-based M&E tools, it's a challenge. But those that using electronic medical records, they can also access their system at ANC, just to register the client right there and then without duplication."*

*(KI 3, Malawi)*

*"Site visits showed us that often some women have two medical files in a same health facility, because they were cared for by one service and when they leave, the new service opens another file for her. When she comes back. . . So, it's almost like she disappears. In fact, it's a traceability issue."*

*(KI 5, Côte d'Ivoire)*

*"So, the clinic where you should go [for ART] is probably, say, two doors out. So, when you leave this room, going there, I don't know whether you decide outside not to go there. . . So, it's possible that probably when the mother leaves the prenatal clinic room, she may decide I am not going to go to the HIV clinic, because I am tired."*

*(KI 1, Malawi)*

Although KIs identified that transition steps could be a vulnerable point for loss from care, quantifying loss over these transition steps is difficult because of limited connection and communication between the various services and no tools to trace whether a woman successfully

transitioned. Reasons the loss is thought to occur include long waiting times at the main ART clinics, as well as women having become accustomed to the ANC providers and being unwilling to go to another site for HIV care. This was thought to be a particularly important issue among women who initiated ART during pregnancy in an integrated service and had never attended an adult ART clinic.

*"We haven't assessed that to say how many loss to follow-ups from those that were in ANC, they have been transferred to their main ART clinic."*

*(KI 3 Malawi)*

*"But we create gaps in care because there's a possibility that a client will decide to just walk away [and not go to the ART clinic]. We may lose clients along the way, during their transfers."*

*(KI 4, Malawi)*

*"The [PMTCT guideline] implementation is average because often the way services are organized cause. . . For instance, monitoring women care. Normally, guidelines state that when a woman tests positive, she must stay within the ANC services until she gives birth and all that. But in some cases, the woman comes and goes."*

*(KI 5*, Côte d'Ivoire*)*

*"That's why I'm saying for the newly identified because maybe the only person she knows is the one she found in the ANC. Of course, you develop a [relationship] or you develop that connection. So when [healthcare worker] hasn't prepared you and then we are like okay, next month you are going to this. Who knows, maybe someone goes back home and says I'm not going there. It's really tricky to explain why the retention is low, especially among the pregnant or breastfeeding women."*

*(KI 5, Malawi)*

## Strategies being used to support transition

Although there is no formal guidance, KIs in all three countries reported that facilities are using strategies to try to support women to transition between HIV service delivery points. Due to the issues with tracing women through these transitions, the success of these strategies has not been assessed.

In the Lesotho ART guidelines, duplicate referral letters are recommended to provide communication when patients are referred between HIV service points including between the maternal and child health services and HIV and ART services. However, KIs felt that further documentation of the number of mothers transferred and numbers received was necessary. KIs reported that a referral is made concrete by providing a date that the woman must attend the new clinic and by following up with a phone call by the provider to inform the receiving clinic; however, there is no requirement in the guidelines for this follow-up to occur.

*"It's good that transfer letter will be a communication to the main ART, and then we also need not from ANC only, but also, the main ART tool, put that to ANC saying we have received—in this month we have received ten mothers who have completed their 24 months, and maybe to get the information from the main ART data, we have referred ten pregnant*

*mothers from ART to ANC that, and then to have a clear documentation data on that which can be reported monthly."*

*(KI 2, Lesotho)*

*"It actually depends on the passion of the service provider to actually make sure she closes the loop."*

*(KI 3, Lesotho)*

KIs in Lesotho also reported that women who are identified as pregnant in ART services are escorted to ANC to join integrated ANC and ART services. In Malawi, KIs reported that in some facilities escorts are used to help to link women from one service to another, allowing women to skip the queue at the main ART clinic.

*"And she [pregnant woman in ART services] is actually physically escorted to that department [ANC] to make sure that she is safely arrived there."*

*(KI 4, Lesotho)*

*"We may lose clients along the way, during their transfers. And that is mitigated by getting an escort from one clinic to another so that they don't queue again, they just go straight to the front and get their medicines and go. But given the type of infrastructure that we have, it varies from one facility to another."*

*(KI 4, Malawi)*

## The role of the community

The role of the community and relationships between the health services and the community were recognized as a very important and underutilized link. Community-based counsellors and peer educators can identify women who are pregnant and counsel them on early presentation for ANC and HIV services. They can also provide support for ongoing retention in PMTCT services, and continued HIV care postpartum. The community will also be critical to the implementation of community-based differentiated service delivery models, but strong links to the health services will be needed to ensure that clinical care providers know which patients are doing well in community services and who requires additional support. Similarly, community counsellors and peer educators could play a valuable role in supporting required transitions of care for PMTCT.

*"If we work on our relationship with communities—our work in communities is structured and there is an efficient relationship—yes, maybe little Boris did not come in, but my colleague can tell me that he is in perfect health and he got his vaccinations thanks to an advanced strategy and he was seen by nurses, I think that is a good model. If a mother can get her ARV in the community. We need to bring healthcare closer to people."*

*(KI 2, Côte d'Ivoire)*

Community-based differentiated care models are already active in Lesotho and Malawi has introduced a package of community PMTCT services. Côte d'Ivoire has not yet implemented this but is considering doing so.

*"When she's done being pregnant and when she is sort of graduated. So there, it's a group of six and it's done in the community. You may find that I would go to the facility like once or twice a year where I would be checked and drawn blood. . .the members of the group rotate. However, on a monthly basis there's one member that goes to the facility and collects drugs for all of us all. And then they also meet in the community and then distribute the drugs to us."*

*(KI 3, Lesotho)*

*"We're trying to take PMTCT services to the community. So, previously, we never had a community PMTCT package. . .So, in the current strategy, we have a community PMTCT package, trying to address issues at community level, and not at the health facility level. And kind of—in the current guideline, we also are trying to make sure that exposed infants are tested right there, and then"*

*(KI 3, Malawi)*

*"We are providing service delivery for women living pretty far from the health facility but who live close to each other. We can gather them together and every month, one of them travels to the health facility for her medical check-up or her viral load test. And while she is picking up ARV drugs for herself, she also picks up the ARV drugs for the other women."*

*(KI 1, Côte d'Ivoire)*

Mentor mothers or expert guides, while not specifically providing transition support, do broadly support women to remain in care during and after pregnancy. Mentor mothers from the community are linked to the facilities and since they have gone through the system themselves, they provide guidance and follow up with women who have missed scheduled visits. Mentor mother programs provide an opportunity to support transition between services and encourage ART retention for the duration of PMTCT.

*"These mothers or mentors or expert counselors, they [stay with them] through the first—the system. And then, of course, ongoing supports because most of these expert counselors, mentor mothers are community-based [from within the community]; they're from the facilities. . .Then just ongoing support like the appointment there. . . Oh, yesterday we were expecting you at the clinic; you didn't come. That's part of the follow up."*

*(KI 5, Malawi)*

*"I am trying to set up pilot studies to find out how we can bring pregnant women who are HIV-positive together—get, what are they called tutoring mothers, to make house calls and visit with community counsellors to get those women to stay in the PMTCT program as long as possible."*

*(KI 3, Côte d'Ivoire)*

Related to this, having partner support was seen as important for women to remain in PMTCT services. Engaging more broadly with the community may make it easier for women to get the care she needs without partner support if necessary, or to engage with her partner about these issues.

*"What can I, as a woman who has not shared her status with her husband, do? We try to work things out with our colleagues in the communities so that every month there are doctors*

*who come to provide care, and everyone comes out. We do not have to go—and it is the part-*
*ner who says, get the child's health records, we are going to see if there are vaccinations they*
*need—the woman, too. If connections are established correctly, they will say today let us go*
*refill our ARV."*

*(KI 2,* Côte d'Ivoire*)*

*"I think partner support is also playing an important role. If the partner is very supportive, the*
*woman will make sure that they get the necessary care. That's my assumption."*

*(KI 4,* Malawi)

## Discussion

All women living with HIV should receive uninterrupted ART for life, including through preg-
nancy and postpartum. Effective healthcare transitions are needed along the PMTCT cascade
for women to continue to access ART without interruptions. These KIIs from three PMTCT
priority countries highlight two key findings: 1) there is a lack of clear guidance on how and
when transitions of care for continued ART should take place, and 2) although transitioning
care is recognized as a potential point of loss along the PMTCT cascade, standardized moni-
toring and evaluation tools do not exist to track transitions of care or rapidly identify women
who do not successfully transition. In addition, several opportunities were identified to sup-
port transitions of care for continued ART along the PMTCT cascade.

The lack of clear guidance on when and how to transition women for continued ART during
PMTCT and how to measure these transitions, along with variable infrastructure and resources
at different health facilities, has resulted in a range of transition approaches. This lack of a stan-
dard approach places pressure on health providers to take the initiative to follow up on women
who transition, rather than having a set protocol and clear responsibilities for the transferring
and receiving clinics. The lack of clarity can also create confusion for women living with HIV if
the reason for transition is not explained and clear instructions around transition are not pro-
vided [10, 24, 25]. In a summary of national PMTCT policies and WHO guidance in Malawi,
South Africa and Tanzania, Jones et al. [11] noted considerable variation in the timing of transi-
tion of women for continued ART during and after pregnancy. The study assessed key indicators
of linkage to care, specifically referral of women to ART clinics from PMTCT, documenting this
referral, having a health worker accompany a woman during the referral and checking if a
woman arrives at the ART clinic from ANC. They found that all three countries had some
guidance on when a woman should be referred from PMTCT to continued ART, but the other
indicators were often not explicitly included in national guidelines. Without guidelines on the
transition approach, it is difficult to define and implement appropriate indicators to document
the occurrence and completion of transition from one service to another. The lack of these indica-
tors, in turn, makes it difficult to fully understand gaps and strengthen these steps in the cascade.

Studies from sub-Saharan Africa and other settings have described the use of transfer let-
ters, transition navigators, case managers, and electronic systems to ensure continuity of care
through required transitions along the PMTCT cascade. As described by KIs in Lesotho, a
recent systematic review found that referral letters are commonly used to support transitions
of care [15]. Referral letters serve as provision of instructions to the woman prior to transition,
and also provide necessary details to the receiving service to support continuity of care. How-
ever, a letter alone, without notification of the receiving service to expect the patient or active
follow up from the referral service to ensure linkage, is not enough to identify when women do

not successfully transition. KIs in Malawi and Lesotho described the use of "escorts" to facilitate transitions of care and both Malawi and Côte d'Ivoire utilize mentor mothers to support women through PMTCT. Variations of transition navigators and mentor mothers for PMTCT are also described widely in the literature [26, 27]. In addition to accompanying women between services to support women remaining in care, they often provide counselling and peer-support for a range of issues relating to maternal and child health, and would be ideally placed to provide transition support. Studies in South Africa and Uganda have shown improved linkage to initiate ART during pregnancy with transition navigators versus transfer letters alone [5, 28]. Selection of optimal transition strategies and timing will also need to consider other contextual issues such as the structure of local services for ANC and HIV as well as family planning services, fertility rates and timing between births.

The formal hand over of patients between services links closely to the need for data tools and indicators to document transitions of care along the PMTCT cascade and identify successful and unsuccessful transitions. This is critical to enable patient-level interventions and timely tracing of women who do not successfully transition. All KIs in this study felt that women were lost through transitions of care but, in most settings, data systems do not allow for easy quantification of this loss. The latest UNAIDS targets include having 95% of pregnant women tested for HIV, all pregnant and breastfeeding women living with HIV diagnosed and on ART, and 95% virally suppressed by delivery and during breastfeeding by 2025 [29]. In order to measure and reach these global targets, programs need to be able to monitor women throughout the PMTCT cascade and targets need to be linked to indicators which ensure continuity of ART during and after pregnancy. A study from Nigeria described how women on ART who become pregnant are allocated a unique PMTCT identifier that is linked to her ART record and, after delivery, her infant's identifier, in order to monitor her completion of the PMTCT cascade even through transitions of care [30]. Case managers and management teams have been successfully implemented in the US, but this resource-intense approach may not be feasible in low-resource, high-burden settings [31, 32]. A study in Myanmar described a similar but more centralized approach through the appointment of a district PMTCT doctor and a PMTCT focal nurse who together were responsible for documenting linkage between ANC and ART services for ANC facilities in a district [33].

In order to link patients transitioning between HIV and maternal and child health service points, electronic medical records using unique patient identifiers linked across facilities will be needed [34]. In India, an electronic line listing was developed to include all women living with HIV presenting for ANC. This list is shared between all clinics to facilitate continuity of care and tracing of women through pregnancy and up to 18 months postpartum [35]. Robust electronic systems that are integrated into routine health information systems and combined with clear responsibilities allocated to the referring and receiving health services, would reduce the risk of women falling through the cracks when transitioning between services. Cohort monitoring through the PMTCT cascade also has the potential to improve continuity of care but is made challenging by transitions of care required for continued ART during and after pregnancy. In addition to these required transitions, women may move between services and geographic areas during and after pregnancy for a range of other personal, social and economic reasons [36]. Thus, investing in unique patient identifiers and systems that have the ability to link women across services in a subdistrict, district or country will be essential [34]. In the absence of these data systems, other communication routes between health services must be improved and the responsibilities of referring and receiving health services clearly defined to ensure no women are lost when transitioning between services for continued ART.

All KIs reflected on the important role that community engagement could play in improving connections between health services. Studies from Mozambique and Tanzania have

described the use of community interventions to actively identify pregnant women in the community. This approach is used to ensure linkage to ANC and PMTCT for all women living with HIV, including women already on ART, and to encourage early ART initiation in pregnancy [37, 38]. Similarly, community-based health services and other differentiated models of care present important opportunities to improve care along the HIV and the PMTCT cascade [3, 39]. The COVID-19 pandemic has further emphasized the importance of scaling up differentiated models of care. KIs in Malawi and Lesotho reported that community-based differentiated models of care are already being implemented and Côte d'Ivoire has plans to implement. It is essential that clinic- and community-based models of ART care also actively identify pregnant and breastfeeding women and provide critical links to ANC and postnatal services. Transitions of care between services, differentiated or not, will be needed, and it is important to note that good links to the existing health services and strong documentation and data systems to track transitions into and out of community interventions and differentiated models of care must not be overlooked.

The following limitations should be noted. Only three of 21 priority countries for PMTCT were included in these KIIs which may limit the generalizability of findings across sub-Saharan Africa. However, very similar themes emerged in all three included countries, despite their diverse geographic, cultural and HIV epidemiologic contexts, and the findings align with results from policy reviews in other countries [11, 40]. While the need for transition guidance and monitoring tools are likely broadly applicable to other countries in the region, the development of specific guidelines for transition approaches and monitoring will need to consider the local context alongside the broad challenges identified in this work. This study also aimed to collect information from key country stakeholders on innovative approaches and pilot projects to support transitions of care for continued ART along the PMTCT cascade. Although stakeholders reported on opportunities for support interventions, no existing projects focused on supporting transitions of care for continued ART along the PMTCT cascade were reported. Only 15 of 33 invited KIs agreed to participate and half were from international partner organizations which may have results in a bias in perspectives towards this group. There is also potential for social desirability bias from all KIs as this work was supported by CDC/PEPFAR who also support country programs. Interviews were conducted with high-level stakeholders involved in national ART and PMTCT programming. As such, this study may not reflect all the experiences, challenges and innovations of specific facilities and in-facility health providers. However, these results provide a big picture perspective on the transition landscape from the perspective of individuals who are responsible for implementing policy and program changes.

These findings provide important insights into improving transition practices and overcoming challenges for policymakers and program implementers. They are also a call to action, in line with priorities to improve maternal retention in HIV care and accelerate progress towards elimination of mother-to-child transmission of HIV [3, 41]. While transition of care for continued ART along the PMTCT cascade is a neglected area, important opportunities exist to develop guidance for these transitions, define indicators to measure transition success, and implement systems to improve monitoring of transitions along the PMTCT cascade. Closing these transition gaps will move us one step closer to global 95-95-95 targets and elimination of mother-to-child transmission of HIV.

## Supporting information

**S1 Appendix. Key informant interview guide.**
(PDF)

## Author Contributions

**Conceptualization:** Chloe A. Teasdale, Amanda Geller, Bernadette Ngeno, Surbhi Modi, Elaine J. Abrams.

**Formal analysis:** Tamsin K. Phillips, Halli Olsen.

**Funding acquisition:** Elaine J. Abrams.

**Project administration:** Halli Olsen, Chloe A. Teasdale, Mamorapeli Ts'oeu, Nicole Buono, Dumbani Kayira.

**Writing – original draft:** Tamsin K. Phillips.

**Writing – review & editing:** Tamsin K. Phillips, Halli Olsen, Chloe A. Teasdale, Amanda Geller, Mamorapeli Ts'oeu, Nicole Buono, Dumbani Kayira, Bernadette Ngeno, Surbhi Modi, Elaine J. Abrams.

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
