## [Decision Letter · Decision Letter 0]

9 Sep 2021

PONE-D-21-23818Uninterrupted HIV treatment for women: policies and practices for care transitions during pregnancy and breastfeeding in Côte d’Ivoire, Lesotho and MalawiPLOS ONE

Dear Dr. Phillips,

Thank you for submitting your manuscript to PLOS ONE. After careful consideration, we feel that it has merit but does not fully meet PLOS ONE’s publication criteria as it currently stands. Therefore, we invite you to submit a revised version of the manuscript that addresses the points raised during the review process.

I agree with the comments and suggestions made for improvement by reviewers 1, 2.

We look forward to receiving your revised manuscript.

Kind regards,

Jodie Dionne-Odom, MD

Academic Editor

PLOS ONE

Journal Requirements:

2. When reporting the results of qualitative research, we suggest consulting the COREQ guidelines: http://intqhc.oxfordjournals.org/content/19/6/349. In this case, please consider including more information on the number of interviewers, their training and characteristics; and please provide the interview guide used.

3. We note that Figure 1 in your submission contain copyrighted images. All PLOS content is published under the Creative Commons Attribution License (CC BY 4.0), which means that the manuscript, images, and Supporting Information files will be freely available online, and any third party is permitted to access, download, copy, distribute, and use these materials in any way, even commercially, with proper attribution. For more information, see our copyright guidelines: http://journals.plos.org/plosone/s/licenses-and-copyright.

a) You may seek permission from the original copyright holder of Figure 1 to publish the content specifically under the CC BY 4.0 license. 

5. Please remove your figures from within your manuscript file, leaving only the individual TIFF/EPS image files, uploaded separately.  These will be automatically included in the reviewers’ PDF.

Reviewers' comments:

Reviewer's Responses to Questions

**Comments to the Author**

1. Is the manuscript technically sound, and do the data support the conclusions?

Reviewer #1: Yes

Reviewer #2: Yes

2. Has the statistical analysis been performed appropriately and rigorously? 

Reviewer #1: Yes

Reviewer #2: Yes

3. Have the authors made all data underlying the findings in their manuscript fully available?

Reviewer #1: No

Reviewer #2: No

4. Is the manuscript presented in an intelligible fashion and written in standard English?

Reviewer #1: Yes

Reviewer #2: Yes

5. Review Comments to the Author

Reviewer #1: The researchers carried out the study to understand perspectives and experiences of stakeholders in three PMTCT priority countries (Malawi, Lesotho and Cote D’Ivoire) regarding the transition of women between PMTCT and ART clinics during and after pregnancy. This was done to inform key policy actors and stakeholders on the existing landscape, challenges, and opportunities for improving transitions and progress towards the elimination of MTCT. They used a qualitative approach in which they conducted key informant interviews with 15 conveniently selected stakeholders from the three countries, five from each country. Study findings were summarized and related to several existing studies on the topic published from Sub-Saharan African. The authors concluded that their study findings provide important insights into the challenges and opportunities for improving the transitions and retaining patients in care, supporting the findings with a lot of quotations from the informants.

I see this as an important and well conducted study whose findings reflect the current situation in many countries and point to a critical gap in the effort’s countries are making to enroll and keep pregnant women living with HIV in care for their health and to prevent MTCT. The publication of these findings will remind and motivate program implementers to implement suggested strategies to fill the gaps. In reviewing the manuscript using the Plos One criterion for publication, author guidelines, the Standards for Reporting Qualitative Research (SRQR) published by Bridget C. O’Brien in 2014 and the Consolidated criteria for reporting qualitative studies (COREQ) published by Allison Tong in 2007, I found some minor gaps in the manuscript that may need revisions. I will strongly recommend the publication of the article once the revisions are made.

Key observations

The section on the study design is well outlined and with clarity. However, it leaves out some basic but important details such as follows;

• There is no mention anywhere in the article that it was a qualitative study. There is also no mention of piloting of the interview guide, the use of repeat interviews to check consistency and the average duration for each interview. There is equally no mention of whether the interviews stopped when saturation was achieved and no idea if the transcripts were shared with the participants for review or validation as stipulated in the COREQ guidelines (Tong, 2007). There is also no mention of where the interviews took place. Including statements on these will enhance the credibility and trustworthiness of the findings.

• There was no mention of any theory or conceptual framework guiding the study and no indication if the study led to the development of any theory. This is usually required for qualitative studies and good to be included (Obrien, 2014).

• It may be important to add more on the characteristics of the researcher, his or her experience on the topic, relationship with participants, their assumptions, and personal interests in the topic. These will increase transparency in the conduct of the study and interpretation of findings.

• A total of 53% of participants were from international partner organizations and since participant selection was done by convenience, there is a possibility of bias which could have been included in the limitations. The limitations were presented but did not include this one.

References

Allison Tong, Peter Sainsbury, Jonathan Craig, Consolidated criteria for reporting qualitative research (COREQ): a 32-item checklist for interviews and focus groups, International Journal for Quality in Health Care, Volume 19, Issue 6, December 2007, Pages 349–357, https://doi.org/10.1093/intqhc/mzm042

O’Brien, Bridget C. PhD; Harris, Ilene B. PhD; Beckman, Thomas J. MD; Reed, Darcy A. MD, MPH; Cook, David A. MD, MHPE Standards for Reporting Qualitative Research, Academic Medicine: September 2014 - Volume 89 - Issue 9 - p 1245-1251 doi: 10.1097/ACM.0000000000000388

Reviewer #2: - Findings from 15 interviews with key stakeholders across 3 countries to explore barriers to successful ART transitions of care during pregnancy and postpartum.

- Small study about a big topic across 3 diverse countries.

- Some interesting, important, grim - but not particularly novel - findings around barriers to continuity of ART care for women before, during, after pregnancy are presented with suggestions for improvements in policy.

-

- Suggestions to improve the manuscript:

- More background in the “settings” section would be helpful for readers not familiar with the 3 countries (e.g., some details about their HIV epidemic/fertility rates/MTCT key indicators.) Also some more reflection in the discussion about how typical these 3 countries might/might not be.

- Would love to see the authors go a little bolder in terms of recommendations. It seems like average fertility rates, family size goals, time between births, access and use of contraception in the local setting should also be factored into how/when/where care transitions occur. Can you make an argument about how pre-conception care / immunizations / contraception / could all perhaps be improved - and thus satisfy other public health metrics - if this problem were sorted out. Any bold ideas to suggest?

- Would be helpful to outline for readers (maybe within Figure 1?) what services take place in the different places/models. E.g. what is provided in ANC that is not part of HIV care for women. Of course, the goal is ART without interruption, but the reason services change during preg and postpartum is bc women have unique needs during these times that cannot be met through simple ART care. This is clearly known to this writing team, but it’s not well articulated in the manuscript.

- Introduction: “successful transition between services….” Would be helpful to define what this means. Or if there is not a clear definition worth pointing that out early.

Methods:

- Current draft missing a description of the process for developing the interview guides, the content explored, the process of piloting, a conceptual framework, etc.

- Details on whether / how coding team worked to include feedback from local / collaborating teams is missing. The section on author contributions reads as though first author (UCT) and 2nd author (ICAP) conducted all analyses. Would be nice to spell this out and if no feedback from relevant country partners would explain why / include in limitations.

- Discussions of limitations of social desirability bias are not included – seems important when research sponsored by CDC/PEPFAR.

RESULTS

- In findings would report how many invite / how many responded / how many agreed to participate.

- Understanding the nature of the interview guide, what was explored in the interview guide, how it was developed will also help inform interpretations of the analysis. e.g., is this the primary paper or is this an additional analysis?

- Can you further explore some of the findings? E.g., why would women just go to ANC to “have belly measured” (pg. 8) and go elsewhere for ARVs? Sounds like the policies are also confusing for patients. How does this work align with what research conducted with pregnant / postpartum people with HIV say? Can you think of creative ways why controversies like what we all experienced with DTG a few years ago would be better managed if women’s care was more longitudinal/ less fragmented? Can you make arguments about cost effectiveness? Cost? In discussion?

- 2nd paragraph only gl mentioned that maybe need to be developed / refined are time to postpartum transition to routine ART. The findings uncovered a lot of additional metrics that seem to need definition. Seems like a great opportunity for the authors to lay out some suggested metrics, solutions. Describing the band-aids being developed (e.g., a country-wide line-listing of preg women in India?!?!) seems like an opportunity to explain why these micro level approaches reflect a lack of uniform guidance and support and offer a way forward.

-

Minor ethics comment:

- It is unclear why CDC would consider this not research? They have ethics approvals from CUMC and the local research committees in the reference countries, so all is in order. It just seems odd that CDC would count this as non-research.

6. PLOS authors have the option to publish the peer review history of their article (what does this mean?). If published, this will include your full peer review and any attached files.

Reviewer #1: **Yes: **Mboh Khan Eveline, MPH, PhD

Reviewer #2: No

---

## [Author Response · Author response to Decision Letter 0]

3 Nov 2021

Editorial comments

We have checked and complied with the style requirements.

2. When reporting the results of qualitative research, we suggest consulting the COREQ guidelines: http://intqhc.oxfordjournals.org/content/19/6/349. In this case, please consider including more information on the number of interviewers, their training and characteristics; and please provide the interview guide used.

Further detail has been added to the methods on the interviewers as well as additional details requested by review 1. The interview guide has been included now in supplementary material.

3. We note that Figure 1 in your submission contain copyrighted images. All PLOS content is published under the Creative Commons Attribution License (CC BY 4.0), which means that the manuscript, images, and Supporting Information files will be freely available online, and any third party is permitted to access, download, copy, distribute, and use these materials in any way, even commercially, with proper attribution. For more information, see our copyright guidelines: http://journals.plos.org/plosone/s/licenses-and-copyright.

a) You may seek permission from the original copyright holder of Figure 1 to publish the content specifically under the CC BY 4.0 license. 

Figure 1 included in this manuscript is currently published under the CC BY 4.0 license in the Journal of the International AIDS Society. The attribution and original copyright has been embedded in the figure file and added to the caption.

The Data Availability statement has been updated as follows: All relevant data are within the manuscript and its Supporting Information files. The full transcripts from this study are not publicly available. Approval for public dissemination of the raw transcripts was not obtained as it may still be possible to identify participants through their transcripts.

5. Please remove your figures from within your manuscript file, leaving only the individual TIFF/EPS image files, uploaded separately. These will be automatically included in the reviewers’ PDF.

Figures have been removed from the manuscript and uploaded separately.

References have been checked.

Reviewer #1: 

1. The researchers carried out the study to understand perspectives and experiences of stakeholders in three PMTCT priority countries (Malawi, Lesotho and Cote D’Ivoire) regarding the transition of women between PMTCT and ART clinics during and after pregnancy. This was done to inform key policy actors and stakeholders on the existing landscape, challenges, and opportunities for improving transitions and progress towards the elimination of MTCT. They used a qualitative approach in which they conducted key informant interviews with 15 conveniently selected stakeholders from the three countries, five from each country. Study findings were summarized and related to several existing studies on the topic published from Sub-Saharan African. The authors concluded that their study findings provide important insights into the challenges and opportunities for improving the transitions and retaining patients in care, supporting the findings with a lot of quotations from the informants.

I see this as an important and well conducted study whose findings reflect the current situation in many countries and point to a critical gap in the effort’s countries are making to enroll and keep pregnant women living with HIV in care for their health and to prevent MTCT. The publication of these findings will remind and motivate program implementers to implement suggested strategies to fill the gaps. In reviewing the manuscript using the Plos One criterion for publication, author guidelines, the Standards for Reporting Qualitative Research (SRQR) published by Bridget C. O’Brien in 2014 and the Consolidated criteria for reporting qualitative studies (COREQ) published by Allison Tong in 2007, I found some minor gaps in the manuscript that may need revisions. I will strongly recommend the publication of the article once the revisions are made.

Key observations

The section on the study design is well outlined and with clarity. However, it leaves out some basic but important details such as follows;

There is no mention anywhere in the article that it was a qualitative study. There is also no mention of piloting of the interview guide, the use of repeat interviews to check consistency and the average duration for each interview. There is equally no mention of whether the interviews stopped when saturation was achieved and no idea if the transcripts were shared with the participants for review or validation as stipulated in the COREQ guidelines (Tong, 2007). There is also no mention of where the interviews took place. Including statements on these will enhance the credibility and trustworthiness of the findings.

Thank you for raising these omissions. Additional details have been added in the manuscript text. Interviews in Malawi and Lesotho were conducted in English by the ICAP-New York project coordinator during country visits. The interviewer had no existing relationship with any of the participants. The Côte d’Ivoire interviews were conducted in French by the ICAP Country Director. The interviewer did know some of the KIs as they encountered each other in their work. Both interviewers had postgraduate training in health research and experience with HIV and PMTCT programs and research. Neither interviewer selected participants as names of stakeholders to contact were provided by the CDC in-country staff. All interviews were conducted face-to-face at a private office at the convenience of the KI. Interviews were on average 60 minutes long.

The interview guide was constructed to ask questions about the transition process based on the literature and experience of the investigators and CDC in-country staff. The interview guide was not piloted prior to use. 

All stakeholders who responded the interview invitation and who were available to be interviewed were included so interviews were not stopped due to saturation. Transcripts were not shared with the participants however the findings were reviewed by CDC partners in each country for review and comment. A single interview was conducted with each KI. Transcripts were not shared with the participants.

2. There was no mention of any theory or conceptual framework guiding the study and no indication if the study led to the development of any theory. This is usually required for qualitative studies and good to be included (Obrien, 2014).

This study aimed to gather information and describe the perspectives of policy makers with the issue of transfer of care for continued ART during and after pregnancy. Due to the descriptive nature of this work and the lack of similar studies at the time the study was designed, no conceptual framework was used to guide the interview guide. The framework for transitions along the PMTCT cascade presented in Figure 1from a recent systematic review conducted by authors on this paper was used to guide the analysis but no other conceptual frameworks were used. The study did not lead to the development of theory. Rather key questions were asked about steps in the PMTCT cascade based on experiences and the existing literature, and the results describe these findings to document the PMTCT landscape in relation to transitions of care. The interview guide has now been included as supplementary material and further description has been added to the methods.

3. It may be important to add more on the characteristics of the researcher, his or her experience on the topic, relationship with participants, their assumptions, and personal interests in the topic. These will increase transparency in the conduct of the study and interpretation of findings.

Thank you for this important suggestions. Additional details on the researchers and interviewers have now been added. All researchers involved have an interest in optimizing continuity of care and completion of all steps in the PMTCT cascade. Interviews were conducted by the ICAP-NY project coordinator (Malawi and Lesotho) and the Côte d’Ivoire ICAP country director. 

4. A total of 53% of participants were from international partner organizations and since participant selection was done by convenience, there is a possibility of bias which could have been included in the limitations. The limitations were presented but did not include this one.

This has been added to the limitations.

Reviewer #2:

Findings from 15 interviews with key stakeholders across 3 countries to explore barriers to successful ART transitions of care during pregnancy and postpartum.

- Small study about a big topic across 3 diverse countries.

- Some interesting, important, grim - but not particularly novel - findings around barriers to continuity of ART care for women before, during, after pregnancy are presented with suggestions for improvements in policy.

1. Suggestions to improve the manuscript:

More background in the “settings” section would be helpful for readers not familiar with the 3 countries (e.g., some details about their HIV epidemic/fertility rates/MTCT key indicators.) Also some more reflection in the discussion about how typical these 3 countries might/might not be.

Thank you for this suggestions. Additional detail has been added to the setting section in the methods as well as consideration of this in the discussion.

2. Would love to see the authors go a little bolder in terms of recommendations. It seems like average fertility rates, family size goals, time between births, access and use of contraception in the local setting should also be factored into how/when/where care transitions occur. Can you make an argument about how pre-conception care / immunizations / contraception / could all perhaps be improved - and thus satisfy other public health metrics - if this problem were sorted out. Any bold ideas to suggest?

These are important considerations and there is still a lot of work to do to determine when and how transitions should occur. Without strengthened systems to monitor these transition points it is difficult to quantify which transition approaches and what timing of transition is optimal in different settings. However, we agree that other contextual factors will need to be considered when developing transition guidance and we have added some comments on this to the discussion.

3. Would be helpful to outline for readers (maybe within Figure 1?) what services take place in the different places/models. E.g. what is provided in ANC that is not part of HIV care for women. Of course, the goal is ART without interruption, but the reason services change during preg and postpartum is bc women have unique needs during these times that cannot be met through simple ART care. This is clearly known to this writing team, but it’s not well articulated in the manuscript.

This has been described further in the text.

4. Introduction: “successful transition between services….” Would be helpful to define what this means. Or if there is not a clear definition worth pointing that out early.

Thank you, this has been added to the introduction.

Methods:

5. Current draft missing a description of the process for developing the interview guides, the content explored, the process of piloting, a conceptual framework, etc.

See response to reviewer 1 on this issue. More detail has been added to the methods. The interview guide is also now included as supplemental material.

6. Details on whether / how coding team worked to include feedback from local / collaborating teams is missing. The section on author contributions reads as though first author (UCT) and 2nd author (ICAP) conducted all analyses. Would be nice to spell this out and if no feedback from relevant country partners would explain why / include in limitations.

This has been clarified in the methods.

7. Discussions of limitations of social desirability bias are not included – seems important when research sponsored by CDC/PEPFAR.

This has been added to the limitations

RESULTS

8. In findings would report how many invite / how many responded / how many agreed to participate.

33 potential KI were identified (14 in Malawi, 7 in Lesotho and 12 in Côte d’Ivoire). Details on the number of potential participants has been included in the results and discussed in the limitations.

9. Understanding the nature of the interview guide, what was explored in the interview guide, how it was developed will also help inform interpretations of the analysis. e.g., is this the primary paper or is this an additional analysis?

Additional detail has been added to the methods and the interview guide has been included as supplementary material. This is the primary paper from this study.

10. Can you further explore some of the findings? E.g., why would women just go to ANC to “have belly measured” (pg. 8) and go elsewhere for ARVs? Sounds like the policies are also confusing for patients. How does this work align with what research conducted with pregnant / postpartum people with HIV say? Can you think of creative ways why controversies like what we all experienced with DTG a few years ago would be better managed if women’s care was more longitudinal/ less fragmented? Can you make arguments about cost effectiveness? Cost? In discussion?

Thank you for these insights. We feel some of these issues are outside of the scope of this paper. We have added comment and reference to studies on the experience of women living with HIV to the discussion. 

11. 2nd paragraph only gl mentioned that maybe need to be developed / refined are time to postpartum transition to routine ART. The findings uncovered a lot of additional metrics that seem to need definition. Seems like a great opportunity for the authors to lay out some suggested metrics, solutions. Describing the band-aids being developed (e.g., a country-wide line-listing of preg women in India?!?!) seems like an opportunity to explain why these micro level approaches reflect a lack of uniform guidance and support and offer a way forward.

Additional thoughts around this have been added to the discussion.

12. Minor ethics comment:

- It is unclear why CDC would consider this not research? They have ethics approvals from CUMC and the local research committees in the reference countries, so all is in order. It just seems odd that CDC would count this as non-research.

Thank you for noting this. The procedure for CDC IRB review is such that only research where the CDC staff are directly involved in the human subjects component of the work is reviewed by the CDC IRB. As CDC staff were not involved in data collection and did not have access to the human subjects data the study was deemed not to require review from the CDC IRB. This work received approval from the Columbia University Medical Center (CUMC) IRB and the following local IRB for each country: (1) National Health Sciences Research Committee (NHSRC) in Malawi, (2) Research and Ethics Committee (REC) in Lesotho; and, (3) Comite National D’Ethique des Sciences de la Vie et de la Sante (CNESVS) in Cote d’Ivoire.

---

## [Editor Report · Decision Letter 1]

12 Nov 2021

Uninterrupted HIV treatment for women: policies and practices for care transitions during pregnancy and breastfeeding in Côte d’Ivoire, Lesotho and Malawi

PONE-D-21-23818R1

Dear Dr. Phillips,

We’re pleased to inform you that your manuscript has been judged scientifically suitable for publication and will be formally accepted for publication once it meets all outstanding technical requirements.

Kind regards,

Jodie Dionne-Odom, MD

Academic Editor

PLOS ONE
---

## [Editor Report · Acceptance letter]

22 Nov 2021

PONE-D-21-23818R1 

Uninterrupted HIV treatment for women: policies and practices for care transitions during pregnancy and breastfeeding in Côte d’Ivoire, Lesotho and Malawi 

Dear Dr. Phillips:

I'm pleased to inform you that your manuscript has been deemed suitable for publication in PLOS ONE. Congratulations! Your manuscript is now with our production department. 

Kind regards, 

on behalf of

Dr. Jodie Dionne-Odom 

Academic Editor

PLOS ONE